# Hypothemycin, a fungal natural product, identifies therapeutic targets in *Trypanosoma brucei*

Mari Nishino[1,2†], Jonathan W Choy[3,4†], Nathan N Gushwa[3†], Juan A Oses-Prieto[5], Kyriacos Koupparis[6], Alma L Burlingame[5], Adam R Renslo[2,4,5], James H McKerrow[2,7], Jack Taunton[7,8*]

[1]Tetrad Graduate Program, University of California, San Francisco, San Francisco, United States; [2]Center for Discovery and Innovation in Parasitic Diseases, University of California, San Francisco, San Francisco, United States; [3]Chemistry and Chemical Biology Graduate Program, University of California, San Francisco, San Francisco, United States; [4]Small Molecule Discovery Center, University of California, San Francisco, San Francisco, United States; [5]Department of Pharmaceutical Chemistry, University of California, San Francisco, San Francisco, United States; [6]Biomedical Sciences Graduate Program, University of California, San Francisco, San Francisco, United States; [7]Department of Cellular and Molecular Pharmacology, University of California, San Francisco, San Francisco, United States; [8]Howard Hughes Medical Institute, University of California, San Francisco, San Francisco, United States

*For correspondence: jack.taunton@ucsf.edu

†These authors contributed equally to this work

Competing interests: The authors declare that no competing interests exist.

**Abstract** Protein kinases are potentially attractive therapeutic targets for neglected parasitic diseases, including African trypanosomiasis caused by the protozoan, *Trypanosoma brucei*. How to prioritize *T. brucei* kinases and quantify their intracellular engagement by small-molecule inhibitors remain unsolved problems. Here, we combine chemoproteomics and RNA interference to interrogate trypanosome kinases bearing a Cys-Asp-Xaa-Gly motif (CDXG kinases). We discovered that hypothemycin, a fungal polyketide previously shown to covalently inactivate a subset of human CDXG kinases, kills *T. brucei* in culture and in infected mice. Quantitative chemoproteomic analysis with a hypothemycin-based probe revealed the relative sensitivity of endogenous CDXG kinases, including *Tb*GSK3short and a previously uncharacterized kinase, *Tb*CLK1. RNAi-mediated knockdown demonstrated that both kinases are essential, but only *Tb*CLK1 is fully engaged by cytotoxic concentrations of hypothemycin in intact cells. Our study identifies *Tb*CLK1 as a therapeutic target for African trypanosomiasis and establishes a new chemoproteomic tool for interrogating CDXG kinases in their native context.

## Introduction

Human African trypanosomiasis, or sleeping sickness, is a debilitating and fatal parasitic disease endemic to sub-Saharan Africa (*Fevre et al., 2008*; *Simarro et al., 2008*). Caused by two subspecies of *Trypanosoma brucei*, the disease begins in the hemolymphatic system and later crosses the blood–brain barrier, resulting in sleep disturbances that deteriorate to coma and death. During the course of infection, the parasites evade the host immune system through periodic switching of the variable surface glycoprotein coat, making vaccination-based approaches unlikely to succeed (*Taylor and Rudenko, 2006*; *Horn and McCulloch, 2010*). Currently, only four chemotherapeutics are approved for sleeping sickness. These therapies suffer from poor oral bioavailability, severe toxicity, and/or emerging resistance. Eflornithine, the only drug with a known mechanism of action, is also the only

**eLife digest** Human African trypanosomiasis—commonly known as sleeping sickness—is a debilitating and potentially fatal tropical disease that is widespread in sub-Saharan Africa. It is caused by the single-celled parasite *Trypanosoma brucei*, which is transmitted to humans by the bite of the tsetse fly. The infection takes its name from the disruption of the circadian clock that occurs early on in the disorder and leads to sleep disturbances. If left untreated, *T. brucei* infection leads to coma, organ failure and death.

Most of the existing pharmaceutical treatments for sleeping sickness were developed more than 50 years ago. However, they are only weakly absorbed into the bloodstream—meaning that high doses must be used—and they lead to unpleasant side effects. Moreover, the *T. brucei* parasite is developing resistance to existing drugs, so further research is needed to identify new therapeutic targets.

One promising option could be the parasite's protein kinases. These enzymes, which add phosphate-based chemical groups to proteins, have a key role in regulating protein function and many of them are already being investigated as therapeutic targets for cancers and autoimmune diseases. *T. brucei* has 182 different kinases, suggesting a wealth of potential new targets. However, many of these are similar to human enzymes, and inhibiting the latter could lead to harmful side effects.

Now, Nishino et al. have produced a synthetic version of a microbially derived kinase inhibitor, called hypothemycin, and have shown that it kills *T. brucei* cells grown in culture. Hypothemycin also killed *T. brucei* in infected mice, completely curing the infection in one third of animals, although high doses of the drug led to side effects. Using a chemical biology approach and quantitative mass spectrometry, Nishino et al. found that the main target of hypothemycin was a previously unknown kinase that is essential for *T. brucei* survival. Although hypothemycin itself is probably unsuitable as a treatment due to its lack of specificity, the work of Nishino et al. suggests that its kinase targets deserve further investigation.

therapeutic for sleeping sickness approved in the last 50 years (***Fairlamb, 2003***). Although a combination of eflornithine and nifurtimox, a drug for American trypanosomiasis (Chagas disease, caused by *T. cruzi*), was introduced in 2009 for *T. brucei* infection, no new drugs are near the clinic (***Priotto et al., 2009***).

Protein kinases have been intensely pursued as therapeutic targets for cancer and autoimmune disease, and more than 15 kinase inhibitors have received FDA approval during the past decade. As in humans, protein phosphorylation in *T. brucei* is an essential regulatory mechanism and plays critical but poorly understood roles in its unique life cycle (***Parsons et al., 2005***; ***Nett et al., 2009***). The vast majority of the 182 protein kinases in *T. brucei* remain poorly characterized, and few have been interrogated in their native cellular context with small-molecule inhibitors. High-throughput screening and medicinal chemistry recently led to the discovery of SCYX-5070, the first reported protein kinase inhibitor with efficacy in a murine model of *T. brucei* infection (***Mercer et al., 2011***). Affinity chromatography with an immobilized derivative suggested that SCYX-5070 binds at least six trypanosome kinases related to human mitogen activated protein kinases (MAPKs) and cyclin-dependent kinases, although the extent to which SCYX-5070 engages these kinases in vivo is not known. Genetic studies in *T. brucei* have established essential roles for orthologs of several human kinases, including cyclin-dependent kinases (***Tu and Wang, 2004***; ***Gourguechon and Wang, 2009***), Aurora kinase (***Tu et al., 2006***), polo-like kinase (***Li et al., 2010***), glycogen synthase kinase-3 (***Ojo et al., 2008***), casein kinase-1 (***Urbaniak, 2009***), and DBF-2-related kinases (***Ma et al., 2010***). High-throughput RNAi studies have implicated additional kinases in *T. brucei* proliferation (***Alsford et al., 2011***; ***Mackey et al., 2011***). Despite the knowledge gained from these studies, it is not clear which *T. brucei* kinases make the best therapeutic targets. Achieving sufficient potency to inhibit essential *T. brucei* kinases, while avoiding related human kinases, poses a significant challenge. Moreover, quantifying intracellular kinase engagement by small-molecule inhibitors in *T. brucei* is an unsolved problem. Pharmacological and chemoproteomic approaches are therefore needed to complement genetic studies in the search for new therapeutic targets (***Moellering and Cravatt, 2012***).

Hypothemycin (**1**, *Figure 1A*) is a polyketide natural product that covalently inhibits a diverse subset of human kinases, including MEK, ERK, PDGFR, VEGFR2, and FLT3 (*Schirmer et al., 2006*; *Winssinger and Barluenga, 2007*). Hypothemycin (*Tanaka et al., 1999*) and related compounds (*Barluenga et al., 2010*) have antitumor activity in mouse xenograft models, and one variant has entered clinical trials (*Kumar et al., 2011*). All hypothemycin-sensitive kinases have a common cysteine immediately preceding the catalytic DXG motif (where X is usually Phe or Leu) (*Schirmer et al., 2006*). A homologous CDXG motif is present in 48 of 518 human protein kinases (*Leproult et al., 2011*). CDXG kinases are functionally diverse, encompassing Tyr, Ser/Thr, and dual-specificity kinases distributed throughout the kinome phylogenetic tree. The nucleophilic thiol of hypothemycin-sensitive CDXG kinases undergoes conjugate addition to the *cis*-enone, as revealed by a crystal structure of the ERK2/hypothemycin complex (*Rastelli et al., 2008*). While the CDXG motif is necessary for potent inhibition by hypothemycin, it does not appear to be sufficient. One-third of human CDXG kinases were unaffected by hypothemycin variants screened at a concentration of 1 µM (*Barluenga et al., 2010*). The selectivity of hypothemycin toward endogenous kinases in living cells remains unknown.

In this study, we exploit a semi-synthetic derivative of hypothemycin to identify potential therapeutic targets in *T. brucei*. We found that hypothemycin is potently trypanocidal, both in cell culture and in mice. To identify its molecular targets, we synthesized the equipotent propargyl-hypothemycin derivative **2** (*Figure 2B*). This affinity probe provided a means to purify and identify covalent kinase targets of hypothemycin. More importantly, probe **2** enabled us to quantify the extent of kinase engagement upon exposure to a defined concentration of hypothemycin. In principle, the workflow described in this study—(1) phenotypic evaluation of mammalian kinase inhibitors, (2) target identification with chemoproteomic tools, (3) genetic knockdown of candidate targets, and (4) correlation of intracellular target engagement with phenotype—can be used to identify and prioritize kinase targets in other therapeutic contexts.

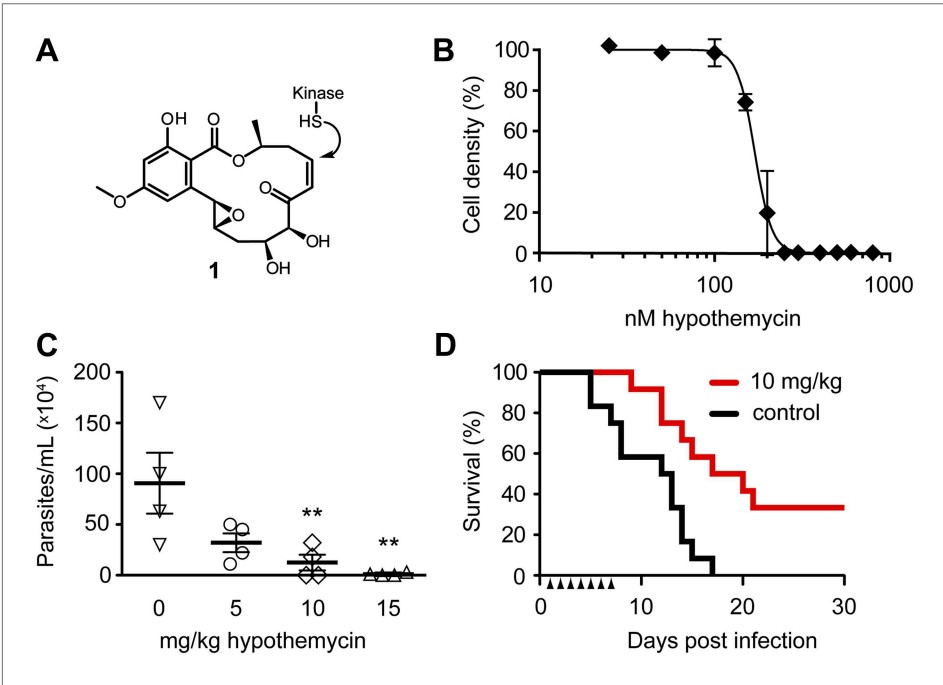

**Figure 1**. Potent trypanocidal activity of hypothemycin. (**A**) Hypothemycin (**1**) inhibits CDXG kinases via conjugate addition to the *cis*-enone. (**B**) Bloodstream form *T. brucei* were treated with hypothemycin and cell density was measured after 24 hr (mean ± SD, n = 3). (**C**) Parasitemia in *T. brucei* infected mice. Mice received once daily intraperitoneal injections of hypothemycin and parasitemia was measured 5 days post infection (mean ± SD, n = 4, ** denotes p<0.05). (**D**) Kaplan-Meier analysis of *T. brucei* infected mice. Hypothemycin or vehicle was administered by intraperitoneal injection once daily for 7 days post infection (arrowheads, n = 12). Data were accumulated from three studies, p<0.01.

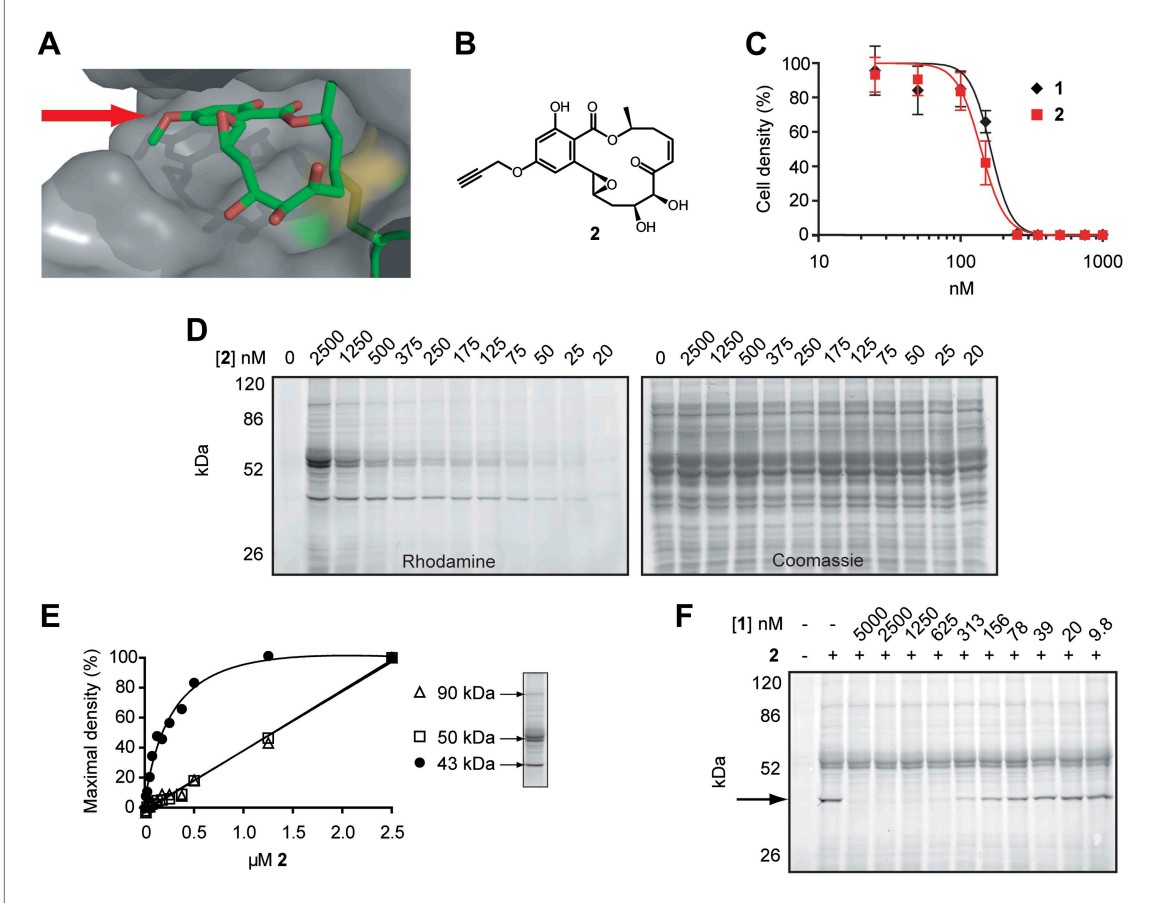

**Figure 2**. Design and validation of a hypothemycin-based affinity probe. (**A**) Crystal structure of hypothemycin (**1**) bound to ERK2 (PDB: 3C9W), indicating the solvent-exposed C4 methyl ether. (**B**) Structure of probe **2**. (**C**) Effect of **2** on proliferation of cultured BSF *T. brucei* (mean ± SD, n = 3). (**D**) *T. brucei* whole-cell lysates were treated with **2** for 30 min. Labeled proteins were visualized after click conjugation to rhodamine-azide and separation by SDS-PAGE. (**E**) Fluorescence quantification of bands at 90, 50, and 43 kDa demonstrating saturation of the 43 kDa band, but not the 90 or 50 kDa bands. (**F**) Lysates were treated with the indicated concentrations **1** for 30 min, followed by 500 nM **2** for 30 min. Labeled proteins were visualized as above. The saturable 43 kDa protein is indicated by the arrow.

## Results

### Hypothemycin has potent trypanocidal activity

Similar to humans, approximately ten percent of *T. brucei* kinases (21 of 182) have a CDXG motif. Hypothemycin's selectivity toward CDXG kinases prompted us to test its effects on *T. brucei* grown in culture. We reasoned that hypothemycin would enable pharmacological interrogation of a focused subset of kinases that are distributed throughout the kinome. Treatment of bloodstream form (BSF) parasites with increasing concentrations of hypothemycin for 24 hr caused a sharp reduction in cell density (EC$_{50}$ ~ 170 nM, *Figure 1B*), principally resulting from cell death. Visual inspection revealed that cells treated for 5 hr with 500 nM hypothemycin were round and swollen, with the flagellum detached from the cell body. They died within 2 hr and did not recover even when transferred to media lacking hypothemycin (data not shown).

Previous studies with hypothemycin demonstrated activity in murine tumor xenograft models (*Tanaka et al., 1999*). We therefore tested hypothemycin in mice infected with *T. brucei*. Infected mice showed a dose-dependent reduction in parasitemia (*Figure 1C*), and a 10 mg/kg dose administered once daily for 7 days prolonged survival of infected mice over 30 days, with a cure rate of 33% (*Figure 1D*). At a dose of 10 mg/kg or higher, signs of toxicity were evident in infected animals (weight loss, lethargy), precluding more aggressive dosing regimens. Although hypothemycin's therapeutic window is narrow, likely due

to inhibition of essential mammalian CDXG kinases (e.g., VEGFR2, MEK1/2), its potent trypanocidal activity in vitro and in vivo motivated us to search for trypanosomal hypothemycin-binding proteins.

## Design and validation of a hypothemycin-based affinity probe

We hypothesized that hypothemycin's trypanocidal effects were mediated by covalent inhibition of one or more *T. brucei* CDXG kinases and sought to identify its targets in an unbiased manner. In the crystal structure of hypothemycin bound to ERK2, the C4 methyl ether is solvent exposed (*Rastelli et al., 2008*), suggesting that this and other kinases would accommodate a larger group at this position (*Figure 2A*). We therefore replaced the methyl ether with a propargyl ether to enable copper-promoted ('click') conjugation to a biotin- or rhodamine-linked azide. Alkylation of 4-*O*-desmethyl hypothemycin (*Wee et al., 2006*) with propargyl bromide was promoted by cesium carbonate to afford the C4 propargyl ether **2**, (*Figure 2B*). Consistent with our hypothesis, probe **2** was equipotent to hypothemycin against cultured BSF *T. brucei* (*Figure 2C*).

We next tested the ability of **2** to covalently modify *T. brucei* proteins. Treatment of whole-cell lysates with **2**, followed by click conjugation to rhodamine-azide, revealed multiple fluorescent bands by SDS-PAGE. Labeling of a 43 kDa protein was detected with 25 nM **2**, reaching maximum intensity at 500 nM (*Figure 2D,E*). Labeled bands at 50 and 90 kDa were apparent at higher concentrations, but they failed to saturate (*Figure 2E*) and are likely nonspecific adducts. Pretreating lysates with increasing concentrations of hypothemycin abolished subsequent labeling of the 43 kDa band, demonstrating that it is a specific and saturable hypothemycin target (*Figure 2F*).

To identify the 43 kDa protein, we used biotin-azide in the click reaction and affinity purified covalently modified proteins. After SDS-PAGE, analysis of the 43 kDa band by mass spectrometry identified the dominant protein (based on total peptide counts) as *Tb*GSK3short (Tb927.10.13780), a CDXG kinase previously found to be essential in BSF *T. brucei* (*Ojo et al., 2008*). Peptides from three other CDXG kinases were also detected in the same gel band (*Supplementary file 1A*). Despite revealing four distinct CDXG kinases, this experiment provided no information about their relative sensitivity to hypothemycin. Moreover, we suspected that abundant and/or hyper-reactive proteins may have bound **2** nonspecifically, obscuring specific, low-abundance targets in the gel-based analysis (*Figure 2D*).

## Hypothemycin sensitivity of CDXG kinases revealed by quantitative mass spectrometry

To identify hypothemycin-binding proteins in a more comprehensive and quantitative manner, we employed a gel-free method with hypothemycin competition and isobaric mass tags, similar to previously reported methods for identifying small-molecule targets (*Bantscheff et al., 2007*; *Huang et al., 2009*; *Ong et al., 2009*). Four lysate samples were treated in parallel with increasing concentrations of hypothemycin for 30 min (0, 20, 200, or 1000 nM), followed by probe **2** (500 nM, 30 min). As before, covalently modified proteins were conjugated to biotin-azide and affinity purified. After trypsinization of the eluted proteins, each sample was derivatized with a unique iTRAQ reagent (Isobaric Tag for Relative and Absolute Quantitation) (*Ross et al., 2004*) and then pooled for fractionation and mass spectrometry analysis. Using this method, peptides corresponding to 10 protein kinases were identified, including three of the four kinases identified by in-gel digest. Four peptides correspond to two nearly identical *Tb*CLK (cdc2-like kinase) genes, *Tb*CLK1 (Tb927.11.12410) and *Tb*CLK2 (Tb927.11.12420), which could not be differentiated on the basis of the identified peptides. In total, we identified 11 potential kinase targets of **2** (*Supplementary file 1A*). Of note, every protein kinase identified by probe **2** contains the CDXG motif, accounting for over half of the 21 CDXG kinases encoded in the *T. brucei* genome. To our knowledge, this represents the first unbiased identification of hypothemycin-binding proteins.

Quantification of iTRAQ reporter ions revealed the extent of CDXG kinase labeling by **2** after pretreatment with increasing concentrations of hypothemycin (*Figure 3*). This experiment thus provides an estimate of each kinase's sensitivity to hypothemycin. Fragmentation spectra of peptides derived from eight kinases provided iTRAQ reporter ions of sufficient intensity to quantify kinase recovery (i.e., labeling by **2**) as a function of hypothemycin pretreatment (*Figure 3B,C*). In addition to CDXG kinases, several other proteins (e.g., tubulin, heat shock proteins, ribosomal proteins) were identified. However, hypothemycin pretreatment did not affect their recovery (*Figure 3B*, *Supplementary file 1A*), which likely resulted from nonspecific, low-level modification by **2** and/or nonspecific adsorption to the avidin-agarose beads.

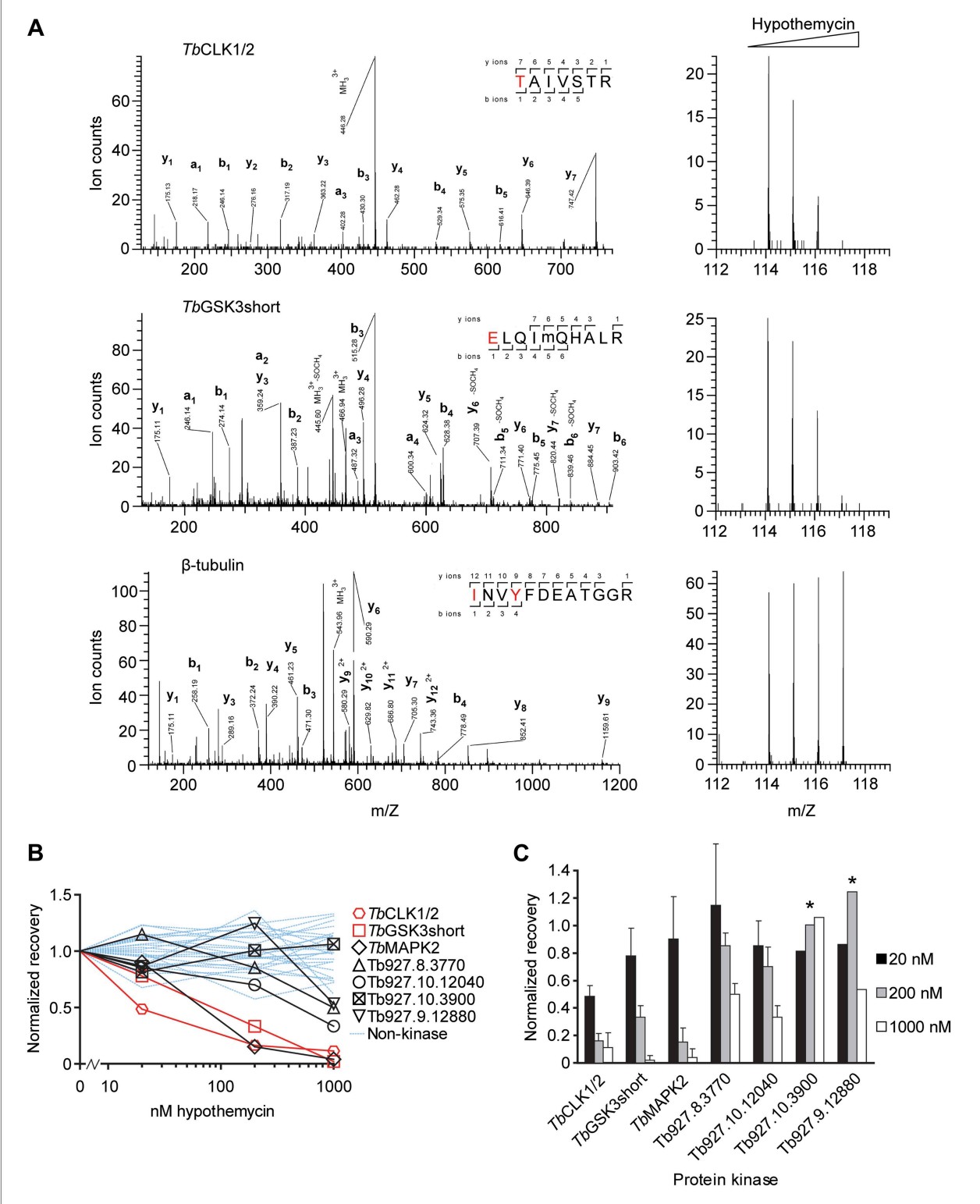

**Figure 3**. Quantification of hypothemycin binding to CDXG kinases. (**A**) MS/MS fragmentation spectra of peptides from *Tb*CLK1/2, *Tb*GSK3short, and β-tubulin after labeling with **2**, affinity purification, and trypsinization. Red text: residues bearing an iTRAQ tag; 'm': oxidized Met. iTRAQ reporter ions for each peptide are shown to the right (m/Z 114, 115, 116, and 117 correspond to lysate samples pretreated with 0, 20, 200, or 1000 nM hypothemycin, respectively). (**B**) Normalized recovery values, based on iTRAQ quantification, for all identified protein kinases and 25 non-kinases (blue lines) with ≥3 unique peptides. (**C**) Recovery of protein kinases as a function of hypothemycin pretreatment. Values for each hypothemycin pretreatment condition are derived from mean iTRAQ reporter ion counts normalized to vehicle across all peptides identified (± SD). Asterisks indicate values derived from a single spectrum.

Among the CDXG kinases quantified, recovery of *Tb*CLK1/2 was uniquely affected by pretreatment with 20 nM hypothemycin (**Figure 3**). Recovery of two additional kinases, *Tb*GSK3short (Tb927.10.13780) and *Tb*MAPK2 (Tb927.10.16030), was reduced more than 50% by 200 nM hypothemycin, and recovery of all but one CDXG kinase was reduced by 1 μM. These data indicate a wide range of hypothemycin sensitivity, with *Tb*CLK1/2 exhibiting somewhat greater sensitivity than *Tb*GSK3short and *Tb*MAPK2. *Tb*MAPK2 is not essential in BSF *T. brucei* but is required for proliferation of the procyclic (insect host) form (**Muller et al., 2002**). *Tb*CLK1 and *Tb*CLK2, which have identical kinase domains and only diverge in their N-terminal regions, are presumed to be equally sensitive to hypothemycin.

## RNAi analysis of CDXG kinases in bloodstream form *T. brucei*

We assessed the requirement for each of the 21 CDXG kinases for cell viability using RNA interference. This required the creation of 21 cell lines, each containing a stably integrated cassette under tetracycline control and designed to silence a unique CDXG kinase (**Wang et al., 2000**). After induction of RNAi, cell proliferation was followed for 6 days. Consistent with previous results (**Ojo et al., 2008**), we observed reduced viability after knockdown of *Tb*GSK3short (**Figure 4A**). Also consistent with previous studies, knockdown of MAP kinases *Tb*MAPK2 (**Muller et al., 2002**), Tb927.6.1780 (**Guttinger et al., 2007**), and Tb927.6.4220 (**Domenicali Pfister et al., 2006**) had no effect on cell viability (**Figure 4B**, **Supplementary file 1B**).

Our initial RNAi construct did not distinguish between *Tb*CLK1 and *Tb*CLK2. Induction with tetracycline resulted in reduced proliferation and increased cell death (data not shown). To test for non-redundant functions of *Tb*CLK1 and *Tb*CLK2, we designed a set of RNAi constructs targeting the unique 5'-UTR of each gene. A lethal phenotype was only observed after knockdown of *Tb*CLK1; knockdown of *Tb*CLK2 had no obvious effect (**Figure 4C,D**). Individual knockdown of the remaining CDXG kinases similarly had no effect on cell proliferation. In each case, knockdown of the corresponding mRNA was confirmed by quantitative PCR (55–80% reduction across 21 mRNAs, **Supplementary file 1B**). With the caveat that insufficient knockdown may underlie the lack of a proliferation defect in certain cases, these results suggest that *Tb*GSK3short and *Tb*CLK1 are the only essential CDXG kinases in BSF *T. brucei*. Remarkably, these kinases also appeared to be among the most sensitive to hypothemycin (**Figure 3**).

## Hypothemycin preferentially targets *Tb*CLK1 in intact parasites

Because both *Tb*GSK3short and *Tb*CLK1 reacted with nanomolar concentrations of hypothemycin in cell lysates and exhibited strong RNAi phenotypes, we focused on these CDXG kinases and assessed

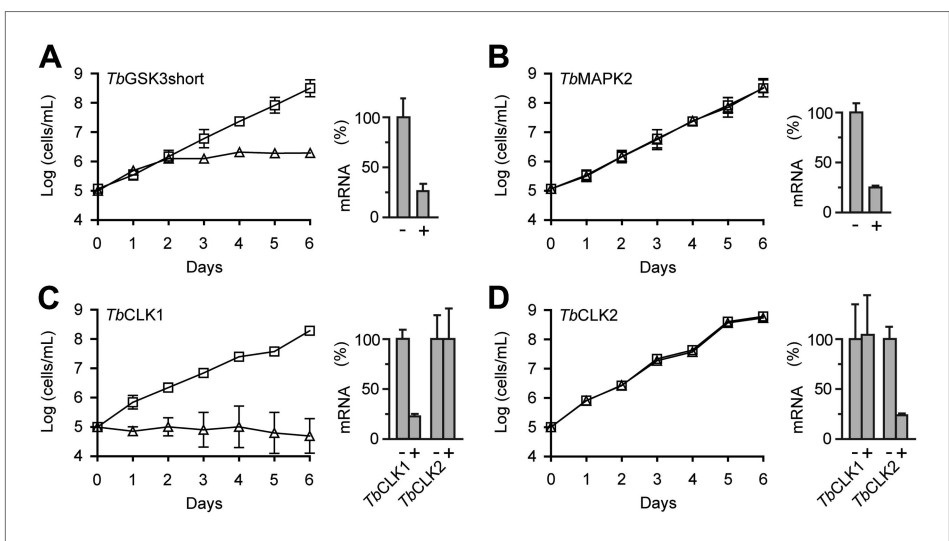

**Figure 4**. RNAi analysis of CDXG kinases. (**A**–**D**) BSF *T. brucei* were stably transfected with the indicated RNAi constructs and induced with tetracycline (triangles) or left uninduced (squares) starting on day 0. Cell density was measured every 24 hr and cumulative cell growth was plotted on a log scale (mean ± SD, n = 3). mRNA levels were measured by quantitative RT-PCR (bar graphs) in the absence (–) and presence (+) of tetracycline.

their sensitivity to hypothemycin in enzymatic assays and in living parasites. Full-length *Tb*GSK3short and *Tb*CLK1 were expressed and purified from *E. coli*. Hypothemycin inhibited *Tb*CLK1 with an $IC_{50}$ of 150 nM when pre-incubated for 30 min in the presence of 100 μM ATP, whereas inhibition of *Tb*GSK3short required 30-fold higher concentrations under the same conditions (**Figure 5A**). This is consistent with our chemoproteomic experiment in which 20 nM hypothemycin reduced subsequent labeling of *Tb*CLK1/2 in lysates, while having little effect on *Tb*GSK3short (**Figure 3**). The greater difference in $IC_{50}$ values in the enzymatic assays is likely due to the presence of competing ATP, which had been depleted from cell lysates prior to the labeling experiments.

The significant difference in hypothemycin sensitivity of recombinant *Tb*GSK3short and *Tb*CLK1, along with their differential sensitivity in lysates, prompted us to ask whether this difference is also observed in intact cells. To quantify hypothemycin binding without mass spectrometry, we generated a strain of BSF *T. brucei* that contains a C-terminal hemagglutinin (HA) tag inserted into the endogenous loci of both *Tb*GSK3short and *Tb*CLK1 (hemizygous for each HA-tagged variant). HA-tagged *Tb*GSK3short and *Tb*CLK1 are easily distinguished on Western blots by their molecular weight. Cells in log-phase growth were treated with increasing concentrations of hypothemycin. After 5 hr, an aliquot of cells from each treatment condition was diluted 30-fold into fresh media, and cell density was measured after 24 hr. The remaining cells were harvested after hypothemycin treatment, and lysates from these cells were treated with 500 nM **2**, followed by click conjugation with biotin-azide. After immunoprecipitation, the HA-tagged kinases were analyzed for covalently attached **2** by Western blot with streptavidin detection.

In control cells, *Tb*CLK1 migrated as two major species (likely corresponding to differentially phosphorylated forms, our unpublished results), which collapsed to a single band upon treatment with hypothemycin (**Figure 5B**). Probe **2** labeled both *Tb*CLK1 bands in the DMSO-treated cells, but labeling was sharply reduced in cells pretreated with hypothemycin. Moreover, the concentration dependence of this effect (reduction of labeling) correlated with the loss of cell viability; the vast majority of cells treated for 5 hr with 500 nM hypothemycin died. In contrast to its potent effect on *Tb*CLK1, hypothemycin had a negligible effect on HA-tagged *Tb*GSK3short, even at the highest concentration tested. Taken together, our results suggest that inhibition of *Tb*CLK1, rather than *Tb*GSK3short, plays a more dominant role in mediating the cytotoxic effects of hypothemycin.

## Discussion

New drugs are urgently needed to treat human African trypanosomiasis. While protein kinases are attractive targets, elucidating which of the 182 *T. brucei* kinases to prioritize for drug discovery efforts

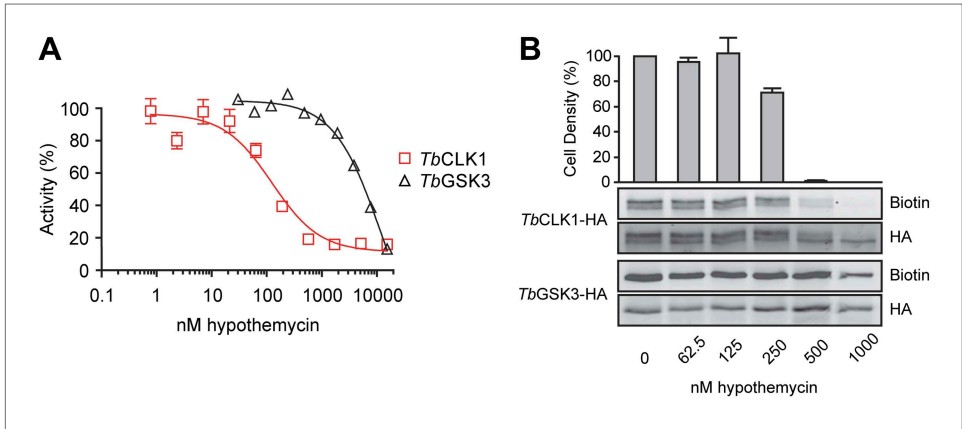

**Figure 5**. Preferential inhibition of TbCLK1 by hypothemycin. (**A**) In vitro assays with recombinant *Tb*GSK3short and *Tb*CLK1. Kinases were incubated with hypothemycin and 100 μM ATP for 30 min before initiating reactions with substrate and $\gamma^{32}$P-ATP. Substrate phosphorylation was quantified and normalized to DMSO control (mean ± SD, n = 3). (**B**) *T. brucei* expressing HA-tagged *Tb*GSK3short and *Tb*CLK1 from their endogenous loci were incubated with hypothemycin for 5 hr, and then were either harvested or diluted 1:30 into drug-free media and counted after 24 hr (bar graph, mean ± SD, n = 3). Harvested cells were lysed, labeled with **2**, and submitted to click conjugation with biotin-azide. HA-tagged proteins were immunoprecipitated, resolved by SDS-PAGE, and analyzed by Western blotting for biotin and HA.

is nontrivial. Many essential *T. brucei* kinases show high sequence identity with human orthologs, frustrating attempts to obtain selective inhibitors. Moreover, once a kinase target has been validated genetically and subjected to high-throughput biochemical screens, demonstrating that a given inhibitor engages the endogenous kinase in intact cells is difficult or impossible; to our knowledge, this has not been accomplished for any *T. brucei* kinase inhibitor. To begin to address these challenges, we have applied a combination of pharmacology, quantitative chemoproteomics, and reverse genetics to the CDXG kinases, a focused yet phylogenetically diverse subset of all eukaryotic kinomes.

We initially discovered that hypothemycin and the propargyl ether derivative **2** are potently trypanocidal. Although hypothemycin reduced parasitemia and cured one-third of infected mice, it was toxic at higher doses, and we do not consider it to be a drug lead. Hypothemycin-mediated toxicity may be caused by inactivation of MEK and VEGFR2, as previous studies have found toxic effects associated with selective inhibitors of these kinases (*Rugo et al., 2005*; *Adjei et al., 2008*). Trypanosomes lack orthologs of most hypothemycin-sensitive mammalian kinases, including MEK and VEGFR2. Conversely, the majority of CDXG kinases in *T. brucei* lack clear human orthologs. It was therefore imperative to develop an unbiased and quantitative method for identifying direct targets of hypothemycin in *T. brucei*.

Of 21 predicted CDXG kinases, 11 were identified by probe **2** after affinity purification from cell lysates. It is likely that the remaining CDXG kinases are not efficiently labeled by **2** or are not expressed in BSF *T. brucei*. Mass spectrometry-based quantification revealed the extent of CDXG kinase occupancy as a function of hypothemycin concentration. We observed a wide spectrum of sensitivity, with most CDXG kinases requiring micromolar concentrations of hypothemycin to block labeling by probe **2**. Two of the kinases showing the greatest reduction in probe labeling after exposure to hypothemycin, *Tb*GSK3short and *Tb*CLK1, are also the only CDXG kinases whose knockdown by RNAi significantly reduced cell growth.

*Tb*GSK3short was previously shown to be essential for proliferation and survival of BSF *T. brucei* (*Ojo et al., 2008*). High-throughput screening with recombinant *Tb*GSK3short has provided inhibitors with high biochemical potency, yet modest selectivity when tested against human GSK3β (*Oduor et al., 2011*; *Urbaniak et al., 2012*), consistent with the high sequence similarity shared by these kinases. Although many of these compounds block *T. brucei* proliferation, it is unclear whether they actually inhibit *Tb*GSK3short in cells. Indeed, hypothemycin was able to saturate *Tb*GSK3short at nanomolar concentrations in cell lysates, yet it had little effect on this kinase in intact cells, even after prolonged treatment at cytotoxic concentrations. Thus, despite *Tb*GSK3short being the most prominent protein labeled by **2** in lysates, it is unlikely that hypothemycin kills *T. brucei* by inhibiting *Tb*GSK3short. Nevertheless, we cannot exclude the possibility that partial inhibition of *Tb*GSK3short or other CDXG kinases contributes to hypothemycin's cytotoxic effects.

Treatment of intact trypanosomes with cytotoxic concentrations of hypothemycin resulted in full occupancy of *Tb*CLK1. Together with the demonstration that *Tb*CLK1 silencing by RNAi is sufficient to kill *T. brucei*, these results validate *Tb*CLK1 as a druggable target. *T. brucei* and other kinetoplastids, including the human pathogens *T. cruzi* and *Leishmania* species, are the only organisms in which a CLK ortholog bears a CDXG motif. The kinase domains of *Tb*CLK1 and human CLK1-4 share only 30% sequence identity. By contrast, *T. brucei* and human GSK3 share 52% sequence identity, including the CDXG motif. Obtaining inhibitors that discriminate between *T. brucei* and human CLK may therefore be more straightforward as compared to GSK3. CLKs have been shown to regulate pre-mRNA splicing in fission yeast (*Tang et al., 2011*), flies (*Du et al., 1998*), and human cells (*Muraki et al., 2004*; *Karlas et al., 2010*; *Fedorov et al., 2011*), and recent studies have revealed important splicing-independent roles (*Rodgers et al., 2010*, *2011*; *Lee et al., 2012*). Further work will be required to elucidate the functions and substrates of *Tb*CLK1 in BSF *T. brucei*.

By employing **2** as an affinity probe, it should now be possible to quantify engagement of both *Tb*GSK3short and *Tb*CLK1 by compounds derived from high-throughput screening campaigns, even without knowledge of their downstream substrates or signaling pathways. We note that *Tb*CLK1 was not among the 57 kinases identified in *T. brucei* cell lysates using broad-spectrum kinase inhibitors immobilized on 'Kinobeads' (*Urbaniak et al., 2012*). Although Kinobeads and other powerful chemoproteomic platforms (*Patricelli et al., 2011*) have the advantage of profiling a larger swath of the kinome, a potential disadvantage is that the probes only work in cell lysates and cannot be used to quantify kinase occupancy in intact parasites. More broadly, probe **2** and the chemoproteomic methodology described herein can be used to explore the related CDXG kinomes of *T. cruzi* and *Leishmania*, as well as uncharted kinomes in other disease-causing eukaryotes, all of which express diverse CDXG kinases. Finally, many human CDXG kinases are validated or potential drug targets (e.g., MEK, ERK, VEGFR2,

c-KIT, FLT3, PDGFR, TAK1, MNK), and probe **2** may prove valuable in quantifying their intracellular engagement by candidate inhibitors.

## Materials and methods

### Cell culture

BSF *T. brucei* (strain 221) was cultured at 37°C and 5% $CO_2$ in HMI-9 medium with 10% Fetal Bovine Serum, 10% Serum Plus (Sigma-Aldrich, St. Louis, MO), 100 units/ml penicillin, and 100 µg/ml streptomycin. The 90-13 cell line was similarly cultured with addition of 2.5 µg/ml G418, 5.0 µg/ml hygromycin. Transgenic cell lines were maintained in medium supplemented with 5.0 µg/ml hygromycin, 2.5 µg/ml phleomycin, and/or 0.1 µg/ml puromycin.

### Cell density assay

The endpoint luciferase-based assay for quantifying cell density was performed as previously described (*Mackey et al., 2011*) with the following modifications. BSF *T. brucei* cells were treated with hypothemycin at $5 \times 10^5$ cells/ml in media lacking antibiotics. Cell density was measured after 24 hr using CellTiter-Glo Luminescent Cell Viability Assay (Promega, Madison, WI). Luminescence was measured using a SpectraFluor Plus luminometer (Tecan, San Jose, CA). Results were plotted using GraphPad Prism (GraphPad Software, La Jolla, CA).

### Mouse infection model

Adult female Balb/c mice (Charles River Laboratories, Wilmington, VA) weighing 18–22 g were infected via intraperitoneal injection with $10^3$ BSF parasites in 100 µl of PBS containing 1% glucose. One day post-infection, hypothemycin (ChemieTek, Indianapolis, IN) in 60% DMSO/water was administered once daily via intraperitoneal injection for 7 days. Mice were monitored every 48 hr for parasitemia in tail blood and visually inspected for general health. Surviving aparasitemic mice at day 30 were considered cured. Experiments were carried out in accordance with protocols approved by the Institutional Animal Care and Use Committee at the University of California, San Francisco.

### Synthesis of probe 2

4-*O*-desmethylhypothemycin (*Wee et al., 2006*) (10 mg, 0.0274 mmol) was added to a dried glass reaction vessel equipped with a magnetic stir bar and dissolved in dry acetonitrile (6 ml). Propargyl bromide (69 µL 80% in toluene, 0.55 mmol) and $Cs_2CO_3$ (10.7 mg, 0.033 mmol) were added. After 6 hr at rt, the mixture was concentrated, dissolved in minimal methylene chloride, and purified by preparative silica TLC using 2% then 3.5% methanol in methylene chloride. Pure **2** (4.5 mg, 41% yield) was eluted from the silica with 10% methanol in methylene chloride. NMR ($^1$H, DMSO, 400 MHz): 0.98 (dd, *J*=14.9 Hz , *J*=9.4 Hz, 1H), 1.36 (d, *J*=6.2 Hz, 3H), 1.86 (dd, *J*=14.0, 9.8 Hz, 1H), 2.55 (m, 1H), 2.76 (m, 1H), 2.94 (dt, *J*=17.3, 10.9 Hz, 1H), 3.61 (t, *J*=2.4 Hz, 1H), 3.87 (m, 1H), 4.32 (d, *J*=1.7 Hz, 1H), 4.45 (dd, *J*=5.1, 1.5 Hz, 1H), 4.83 (d, *J*=2.4 Hz, 2H), 4.93 (d, *J*=5.1 Hz, 1H), 5.15 (d, *J*=6.6 Hz, 1H), 5.39 (m, 1H), 6.11 (dt, *J*=11.2, 2.6 Hz, 1H), 6.32 (d, *J*=2.8 Hz, 1H), 6.42 (dd, *J*=11.7, 2.7 Hz, 1H), 6.51 (d, *J*=2.8 Hz, 1H), 11.90 (s, 1H). NMR ($^{13}$C, DMSO, 400 MHz): 21.02, 34.24, 36.77, 56.39, 57.29, 63.72, 69.54, 74.83, 79.12, 79.51, 81.84, 102.37, 104.00, 105.66, 128.59, 143.10, 143.21, 162.82, 165.09, 171.00, 201.85. HRMS: predicted [M+H+] m/z 403.1387; measured 403.1383.

### Preparation of lysates

BSF *T. brucei* at a density of $10^6$—$5 \times 10^6$ cells/ml were collected and washed twice with PBS. The pellet was suspended in PBS containing Complete Protease Inhibitor Cocktail (Roche, Basel, Switzerland) and PhosStop Phosphatase Inhibitor Cocktail (Roche) and sonicated on ice. Cellular debris was pelleted, and the supernatant was passed through a NAP-5 column (GE Healthcare, Buckinghamshire, United Kingdom) equilibrated with lysis buffer.

### Labeling with probe 2 and click chemistry

Whole-cell lysates (18.75 µl, 1–3 mg/ml protein) were treated with increasing concentrations of hypothemycin for 30 min, followed by the indicated concentration of **2** for 30 min. Samples were denatured with SDS (1.25 µl, 20%), followed by addition of TAMRA-azide (0.5 µl, 5 mM), TCEP (0.5 µl, 50 mM, pH ~7.0), TBTA ligand in 1:4 DMSO:*tert*-butyl alcohol (1.5 µl, 1.67 mM), and $CuSO_4$ (0.5 µl, 50 mM). Reactions were incubated at rt for 1 hr, resolved by SDS-PAGE, scanned for fluorescence (Typhoon Imaging System, Molecular Dynamics, Sunnyvale, CA), and stained with Coomassie.

## Preparation of iTRAQ mass spectrometry samples

*T. brucei* whole-cell lysates (14 mg, 3 mg/ml) were treated with 0 (DMSO control), 20, 200, or 1000 nM hypothemycin for 30 min, followed by **2** (500 nM, 30 min). Samples were then denatured by adding 20% SDS to a final concentration of 1%. Base-cleavable biotin-azide (*Choy et al., 2013*) was added (100 µM), followed by the remaining click reagents at the concentrations described above. After 1 hr at rt, proteins were precipitated with cold acetone (80% vol/vol final). Protein pellets were washed three times with acetone and dried. Pellets were resuspended in a minimal volume of 1% SDS in 50 mM Tris pH 8.0. The solution was then diluted with 9 volumes of 1% NP-40 in PBS and passed through a PD-10 column (GE Healthcare), eluting with 1% NP-40 and 0.1% SDS in PBS. Avidin-agarose (30 µl, Sigma-Aldrich) was added and the samples were rotated overnight at 4°C. The beads were washed twice for 1 hr at rt with 1% NP-40, 0.1% SDS in PBS; twice for 1 hr at 4°C with 6 M urea in PBS; 1 hr with PBS at 4°C; and 1 hr with PBS at rt. To facilitate the quantitative release of captured proteins, the ester linkage (*Choy et al., 2013*) was hydrolyzed with NaOH (0.4 N, 20 µl). After 20 min at room temperature, the solution was neutralized with HCl (0.8 N, 10 µl), brought to 1% SDS, and heated to 90°C for 3 min before collecting the supernatant.

Eluted proteins were acetone-precipitated and pellets were washed twice with additional cold acetone. Dried pellets were resuspended in 8 M guanidinium HCl, reduced (5 mM TCEP, 50 mM ammonium bicarbonate, 6 M guanidinium HCl), and alkylated with 10 mM iodoacetamide. The solution was adjusted to 1 M guanidinium HCl, 100 mM ammonium bicarbonate, 10% acetonitrile and trypsinized overnight. Volatiles were removed, and the samples were resuspended in 0.1% formic acid, extracted with C18 OMIXtips (Varian, Palo Alto, CA), and eluted with 50% acetonitrile, 0.1% formic acid. Volatiles were removed and peptides were resuspended in 0.5 M triethylammonium bicarbonate pH 8.5 and labeled with iTRAQ Reagents (Applied Biosystems, Foster City, CA). Samples were mixed, dried, and resuspended in 30% acetonitrile, 5 mM potassium phosphate (pH 2.7). The samples were then fractionated on a 50.0 × 1.0 mm 5 µM 200 Å Polysulfoethyl A column (PolyLC, Colombia, MD) with a 1–40% gradient of 350 mM potassium chloride in 30% acetonitrile, 5 mM potassium phosphate (pH 2.7). Peptide-containing fractions were dried, resuspended in 0.1% formic acid, extracted using C18 ZipTips (Millipore, Billerica, MA), eluted with 50% acetonitrile, dried, resuspended in 0.1% formic acid, and analyzed by LC/MS/MS.

## Mass spectrometry

Fractionated tryptic peptides were separated by nano-flow liquid chromatography using a 75 µm × 150 mm reverse phase C18 PepMap column (Dionex-LC-Packings, Sunnyvale, CA) at a flow rate of 350 nl/min in a NanoLC-1D Proteomics high-performance liquid chromatography system (Eksigent Technologies, Dublin, CA) equipped with a FAMOS autosampler (Dionex-LC-Packings). Peptides were eluted using a 2–30% gradient of acetonitrile/0.1% formic acid over 40 min, followed by 50% acetonitrile for 3 min. The eluate was coupled to a microionspray source attached to a QSTAR Elite mass spectrometer (Applied Biosystems/MDS Sciex, Framingham, MA). Peptides were analyzed in positive ion mode. MS spectra were acquired for 1 sec in the m/z range between 350 and 1500. MS acquisitions were followed by 4 × 2.5 sec collision-induced dissociation (CID) experiments in information-dependent acquisition mode. For each MS spectrum, the two most intense multiple charged peaks over a threshold of 25 counts were selected for generation of CID mass spectra. Two MS/MS spectra of each were acquired, first on the *m/z* range 119–1500, with Q1 resolution set at 'low', and then on the *m/z* range 112–119, with resolution set at 'unit'. The CID collision energy was automatically set according to mass to charge (*m/z*) ratio and charge state of the precursor ion. A dynamic exclusion window was applied which prevented the same *m/z* from being selected for 1 min after its acquisition.

## Mass spectrometry data analysis

Peak lists were generated using the mascot.dll script. The peak list was searched against the Trypanosoma subset of the NCBInr database using ProteinProspector. A minimal ProteinProspector protein score of 15, a peptide score of 15, a maximum expectation value of 0.1, and a minimal discriminant score threshold of 0.0 were used for initial identification criteria. iTRAQ modification of the amino terminus or the epsilon-amino group of lysines, carbamidomethylation of cysteine; acetylation of the N-terminus of the protein and oxidation of methionine were allowed as variable modifications. Peptide tolerance in searches was 100 ppm for precursor and 0.2 Da for product ions, respectively. Peptides containing two miscleavages were allowed. The number of modifications was limited to two per peptide.

Quantification was based on the relative areas of the reporter ions ($m/z$ =114, 115, 116 and 117) generated by the isobaric iTRAQ reagents during CID experiments. Abundance ratios of individual peptides between the different samples were calculated taking as reference the DMSO-treated sample, by dividing the areas of their respective iTRAQ reporter ions. Peptides with peak areas lower than 30 for the most intense reporter ion were discarded. For changes in relative abundance at the protein level, all MS/MS spectra for the different peptides belonging to a particular protein were used to calculate the average and SD of the abundance ratios.

### Generation of transgenic *T. brucei* cell lines

Plasmid DNA (10 µg) was linearized (MfeI, *Tb*GSK3short-HA; Bsu36I, *Tb*CLK1-HA; NotI, pZJM constructs) and resuspended to 1 µg/µl in water. Cells (2 × 10^7) were electroporated using the Human T Cell Nucleofector kit and the Amaxa program X-001 (Lonza, Basel, Switzerland), then diluted in media and allowed to recover for 24 hr before selection for 5–7 days with the appropriate antibiotic.

### RNA interference

A 250–500 bp fragment of each CDXG kinase gene was amplified from genomic DNA by PCR (primers listed in *Supplementary file 1C*) and cloned into the XhoI/HindIII site of pZJM (*Wang et al., 2000*). Plasmid DNA was stably transfected into BSF *T. brucei* 90–13 cells as described above. RNAi was induced with 1 µg/ml tetracycline. Parasites were counted every 24 hr using a Coulter Counter (Beckman Coulter, Brea, CA), and cultures were diluted to maintain cell density between $10^5$ and $2 \times 10^6$ cells/ml. RNA was extracted 48 hr after tetracycline induction using TRIzol (Invitrogen) and RNeasy kit (Qiagen, Hilden, Germany). For qRT-PCR quantification (relative to *Tb*GAPDH, Tb927.6.4300), cDNA was generated with the High Capacity RNA-to-cDNA Kit (AppliedBiosystems) and quantified using the PowerSYBR kit (AppliedBiosystems).

### In vitro kinase assays

Full-length *Tb*GSK3short and *Tb*CLK1 were amplified from genomic DNA using primers listed in *Supplementary file 1C* and cloned into the pET100 expression vector (Invitrogen). Proteins were expressed in *E. coli* ArcticExpress (DE3) (Stratagene, La Jolla, CA) in ZY5052 medium at 20°C for 60 hr with agitation. Cells were lysed using a microfluidizer into lysis buffer (50 mM Tris pH 8, 300 mM NaCl, 10 mM imidazole, 5% glycerol, 1 mM CaCl$_2$, 1 mM MgCl$_2$, 500 nM PMSF, 1× Protease inhibitor cocktail (Roche), DNase (Sigma-Aldrich), lysozyme (Sigma-Aldrich). The soluble fraction was isolated by centrifugation and incubated overnight at 4°C with Ni-NTA beads. The recombinant protein was eluted with lysis buffer containing 250 mM imidazole and dialyzed into storage buffer (30 mM Tris pH 7.5, 50 mM NaCl, 50% glycerol, 5 mM DTT, 1 mM EDTA, 0.03% Brij35). Aliquots of the proteins were flash frozen with liquid nitrogen and stored at –80°C.

Kinases (5 nM *Tb*GSK3short, 10 nM *Tb*CLK1) were incubated with hypothemycin in reaction buffer (50 mM Tris pH7.4, 10 mM MgCl$_2$, 0.2 mM EGTA, 0.2 mg/ml BSA, 1 mM DTT) and 100 µM ATP for 30 min at rt. A solution of reaction buffer with γ$^{32}$P-ATP (70–150 µCi/ml, Perkin Elmer, Waltham, MA) and GSM peptide substrate (*Tb*GSK3short, 0.05 mg/ml, Millipore) or myelin basic protein (*Tb*CLK1, 0.5 mg/ml, Sigma-Aldrich) was added to initiate the kinase reaction. After 15 (*Tb*GSK3short) or 30 min (*Tb*CLK1), reactions were spotted onto phosphocellulose paper, washed once with 10% acetic acid, twice with 1% phosphoric acid, and once with methanol, and dried. Kinase activity was quantified using a Typhoon Imaging System (Molecular Dynamics) and ImageQuant 5.2 (Molecular Dynamics). IC$_{50}$ values and dosing curves were generated using GraphPad Prism 5 (GraphPad Software).

### Cellular labeling assays

*Tb*GSK3short or *Tb*CLK1 lacking start and stop codons were amplified from genomic DNA using primers listed in *Supplementary file 1C* and inserted between the KpnI and XhoI sites of pC-PTP-PURO, which was modified to contain a C-terminal HA-tag (*Schimanski et al., 2005*; *Gourguechon and Wang, 2009*). For the *Tb*CLK1-HA construct, the puromycin resistance cassette was replaced with a hygromycin resistance gene. *T.brucei* 221 cells were first transfected with the *Tb*GSK3short-HA construct to establish a puromycin resistant cell line in which the vector stably integrated into the endogenous *Tb*GSK3short locus via homologous recombination. This cell line was then transfected with the *Tb*CLK1-HA construct and selected for hygromycin and puromycin resistance. Cells stably expressing HA-tagged *Tb*GSK3short and *Tb*CLK1 (5 × 10^6 cells/ml) were treated with the indicated concentrations of hypothemycin. After 5 hr, aliquots of the cultures were diluted 30-fold with fresh media and cell density was quantified after an additional 24 hr. The remaining cells were pelleted, washed with trypanosome dilution buffer (5 mM KCl, 80 mM NaCl, 1 mM

MgSO$_4$, 20 mM glucose, 22 mM sodium phosphate pH 7.7) and lysed by sonication in 50 μl PBS containing 0.25% NP-40, 1× Complete EDTA-free protease inhibitor cocktail, and 1× PhosStop phosphatase inhibitors (Roche). Lysates were clarified by centrifugation, normalized for protein content, and treated with 500 nM **2** for 30 min. Samples were subjected to click conjugation with biotin-azide as described above and diluted 10-fold with 1.1% NP-40 in PBS. HA-tagged proteins were immunoprecipitated using 12CA5 anti-HA antibody (Roche) and Protein A Dynabeads (Invitrogen), eluted with sample buffer, resolved by SDS-PAGE, and transferred to nitrocellulose membranes. HA and biotin were detected with anti-HA (1:1000, Sigma Aldrich, H6908), IRDye(680)-conjugated anti-rabbit (1:10,000), IRDye(800)-conjugated streptavidin (1:20,000) using the Odyssey Imaging System (Li-Cor Biosciences, Lincoln, NE).

## Acknowledgements

We thank D Santi for 4-O-desmethylhypothemycin, M Abdulla for assistance with mouse studies, and Z Mackey for discussion. Mass spectrometry analysis was provided by the Bio-Organic Biomedical Mass Spectrometry Resource at UCSF, supported by the Biomedical Research Technology Program, NIH NIGMS 8P41GM103481 (ALB).

## Additional information

### Funding

| Funder | Grant reference number | Author |
| --- | --- | --- |
| Howard Hughes Medical Institute | | Jack Taunton |
| National Institutes of Health | GM071434 | Jack Taunton |
| Sandler Foundation | | Nathan N Gushwa, Adam R Renslo, James H McKerrow |
| National Science Foundation | | Mari Nishino, Kyriacos Koupparis |
| Achievement Rewards for College Students Wells Fargo Scholarship | | Jonathan W Choy |
| Genentech Predoctoral Fellowship | | Jonathan W Choy |
| National Institutes of Health | NIGMS 8P41 GM103481 | Alma L Burlingame |

The funders had no role in study design, data collection and interpretation, or the decision to submit the work for publication.

### Author contributions

MN, JWC, NNG, Conception and design, Acquisition of data, Analysis and interpretation of data, Drafting or revising the article; JAO-P, Conception and design, Acquisition of data, Analysis and interpretation of data, Drafting or revising the article; KK, Acquisition of data, Analysis and interpretation of data; ALB, ARR, Analysis and interpretation of data, Drafting or revising the article; JHM, JT, Conception and design, Analysis and interpretation of data, Drafting or revising the article

### Ethics

Animal experimentation: Experiments with mice were carried out in accordance with the Institutional Animal Care and Use Committee at the University of California, San Francisco (IACUC protocol AN087316-02A).

## Additional files

### Supplementary files

• Supplementary file 1. (**A**) Mass Spectrometry Results. (**B**) Summary of RNAi results. (**C**) Primers used in this study.

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
