## [Decision Letter]

Thank you for sending your work entitled “Therapeutic targets in *Trypanosoma brucei* revealed by hypothemycin-based chemoproteomics and RNAi” for consideration at *eLife*. Your article has been favorably evaluated by a Senior editor and 3 reviewers, one of whom is a member of our Board of Reviewing Editors.

The following individuals responsible for the peer review of your submission want to reveal their identity: Jon Clardy (Reviewing editor), Nathanael Gray (peer reviewer), and Barbara Burleigh (peer reviewer).

The Reviewing editor and the other reviewers discussed their comments before we reached this decision, and the Reviewing editor has assembled the following comments to help you prepare a revised submission.

This paper identifies a trypanosome kinase, *Tb*CLK1, as a therapeutic target for African trypanosomiasis, a protozoal infection with severely limited treatment options. The authors used a virtuosic combination of phenotypic assays, organic chemistry, several flavors of proteomics, and RNAi knockdowns in their identification.

The project began by exploring whether kinase inhibitors, which have become useful therapeutic agents for several human diseases, could play a similar role in treating trypanosomal infections. Trypanosomes and their human hosts have many kinases, and to focus their search the authors selected hypothemycin, a small molecule produced by some fungi, that was known to target a small subset of human kinases, and found that it killed trypanosomes in both cell culture and in vivo (mouse) assays. They then synthesized an affinity reagent and through proteomic analysis identified eleven potential targets, all of which were kinases with the signature motif characteristic of hypothemycin-sensitive kinases. More detailed analyses identified one of the kinases, *Tb*CLK1, as the therapeutically relevant target, and this conclusion was underscored with RNAi analyses.

All of the reviewers agreed that the manuscript described the identification of a potentially significant target and were in favor of publication. They also noted that the manuscript describes an unusually thorough set of experiments in a commendably frank manner. Their study will appeal to both the community interested in addressing protozoal diseases like trypanosomiasis as well as the much larger community of researchers seeking to discover useful therapeutic agents. There were two general issues that the reviewers felt could be improved.

1) The authors note in several places that the critical issue in their overall approach is likely to be specificity – the ability to find small molecules that selectively lead to parasite death. The lack of sufficient specificity is, for example, why the authors note that hypothemycin is not a drug candidate. It would be useful to consolidate and expand upon this notion in the Introduction so that the reader is aware of it from the beginning. For example, the Abstract states that “Protein kinases are attractive therapeutic…”, and changing “are” to “could be” would set up such a discussion. The authors do address this in the beginning of the Discussion section, but it might be more effective to move this into the Introduction.

2) The thoroughness of the approach makes it difficult for the majority of readers that are unfamiliar with this sort of analysis to follow the logic of the overall research plan. A road map of phenotypic screen of a mammalian kinase inhibitor -> target ID by chemical proteomics in gel -> chemical proteomics using ITRAQ -> RNAi screen of candidate targets -> target engagement studies -> reconfirmation with recombinant enzymes, possibly as a figure, would help. It would serve both as an aid for readers and a handy guide for repurposing other kinase inhibitors for infectious disease applications. The current manuscript seems to follow the historical course of the research, which creates a meandering narrative.

---

## [Author Response]

*1) The authors note in several places that the critical issue in their overall approach is likely to be specificity – the ability to find small molecules that selectively lead to parasite death. The lack of sufficient specificity is, for example, why the authors note that hypothemycin is not a drug candidate. It would be useful to consolidate and expand upon this notion in the Introduction so that the reader is aware of it from the beginning. For example, the Abstract states that “Protein kinases are attractive therapeutic…”, and changing “are” to “could be” would set up such a discussion. The authors do address this in the beginning of the Discussion section, but it might be more effective to move this into the Introduction*.

These are good suggestions. We have changed the Abstract to indicate that protein kinases are “potentially attractive” therapeutic targets. In addition, we have expanded the introductory paragraph on kinases to include a brief discussion on the challenges associated with developing selective kinase inhibitors and choosing specific kinases to pursue as therapeutic targets.

*2) The thoroughness of the approach makes it difficult for the majority of readers that are unfamiliar with this sort of analysis to follow the logic of the overall research plan. A road map of phenotypic screen of a mammalian kinase inhibitor -> target ID by chemical proteomics in gel -> chemical proteomics using ITRAQ -> RNAi screen of candidate targets -> target engagement studies -> reconfirmation with recombinant enzymes, possibly as a figure, would help. It would serve both as an aid for readers and a handy guide for repurposing other kinase inhibitors for infectious disease applications. The current manuscript seems to follow the historical course of the research, which creates a meandering narrative*.

Although hypothemycin’s effects in the mouse infection model are likely complicated by its interaction with host kinases, we chose to focus on trypanosome targets due to the dramatic effects on the free bloodstream form parasites in culture. We have changed the text to clarify our rationale for pursuing trypanosome targets. As mentioned in the main text, hypothemycin and derivatives have been previously used in mice (41; 4) and in humans (16). We are unaware of any hemolytic activity of hypothemycin or its derivatives.